# Non-Recombinogenic Functions of Rad51, BRCA2, and Rad52 in DNA Damage Tolerance

**DOI:** 10.3390/genes12101550

**Published:** 2021-09-29

**Authors:** Félix Prado

**Affiliations:** Centro Andaluz de Biología Molecular y Medicina Regenerativa (CABIMER), Consejo Superior de Investigaciones Científicas, Universidad de Sevilla, Universidad Pablo de Olavide, 41092 Seville, Spain; felix.prado@cabimer.es

**Keywords:** homologous recombination, Rad51, BRCA2, Rad52, DNA damage tolerance

## Abstract

The DNA damage tolerance (DDT) response is aimed to timely and safely complete DNA replication by facilitating the advance of replication forks through blocking lesions. This process is associated with an accumulation of single-strand DNA (ssDNA), both at the fork and behind the fork. Lesion bypass and ssDNA filling can be performed by translation synthesis (TLS) and template switching mechanisms. TLS uses low-fidelity polymerases to incorporate a dNTP opposite the blocking lesion, whereas template switching uses a Rad51/ssDNA nucleofilament and the sister chromatid to bypass the lesion. Rad51 is loaded at this nucleofilament by two mediator proteins, BRCA2 and Rad52, and these three factors are critical for homologous recombination (HR). Here, we review recent advances showing that Rad51, BRCA2, and Rad52 perform some of these functions through mechanisms that do not require the strand exchange activity of Rad51: the formation and protection of reversed fork structures aimed to bypass blocking lesions, and the promotion of TLS. These findings point to the central HR proteins as potential molecular switches in the choice of the mechanism of DDT.

## 1. Introduction

Homologous recombination (HR) mechanisms repair DNA breaks using intact homologous DNA sequences as template (donor molecule). If the template contains some heterology, the recombination process might be associated with a transfer of genetic information, otherwise parental and recombinant products will be identical (e.g., sister chromatin recombination). The search for homology and strand exchange are central molecular steps for HR mechanisms associated with both double-strand break (DSB) repair and single-strand DNA (ssDNA) filling. These steps are carried out by a ssDNA molecule coated with the Rad51 protein (RecA in bacteria), which is the central protein in HR. Even though a number of additional factors participate in the dynamics of Rad51 on this nucleofilament, two are critical for Rad51 loading: BRCA2 in mammalian cells and Rad52 in yeast. The organisms where HR mechanisms have been studied in more details [1]. The analyses of stalled replication forks by different stress conditions revealed important non-recombinogenic functions (defined as those that do not require the strand exchange activity of Rad51) of these factors in the dynamics of reversed fork structures [2,3]. The number of non-recombinogenic functions of Rad51 and Rad52 have been recently extended to translesion synthesis (TLS) [4,5,6], an error-prone mechanism of DNA damage tolerance (DDT) in which specialized polymerases incorporate a dNTP opposite a lesion that blocks DNA replication [7].

## 2. DSB Repair by HR

Most of our knowledge about the mechanisms and factors involved in HR comes from the study of DSB repair both in mitosis and meiosis, where the recombinational repair of programmed DSBs is critical for the formation of viable meiotic products. Indeed, for a long time, the fact that DSBs induce HR led to the assumption that hyper-recombination in mutant backgrounds, or in response to genotoxic agents, was always mediated by a DSB. Genetic and molecular studies on DSB repair, together with the biochemical characterization of the factors required for HR, allowed to outline the main HR mechanisms [1,8]. They all start with the resection by specialized DNA nucleases of the 5′-ends of the DSB to generate 3′-ended ssDNA molecules (Figure 1). These nucleases include the MRX/N complex—formed by Mre11, Rad50 and Xrs2 (yeast), or Nbs1 (mammal), Sae2 (yeast)/CtIP (mammal), Exo1, and Dna2, which works together with the helicase Sgs1 (yeast)/BLM (mammal) [9]. The 3′-ended ssDNA molecule is covered and protected by the ssDNA binding heterotrimeric complex RPA. BRCA2 and Rad52 promote the loading of Rad51, thus displacing RPA and generating a ssDNA/Rad51 nucleofilament. BRCA2 and Rad52 are helped in this task by accessory factors: Rad55, Rad57, and the Shu complex in yeast; RAD51B, RAD51C, RAD51D, XRCC2, and XRCC3 in mammals; and Rad54 in yeast and mammals.

Rad51 forms onto the ssDNA molecule a right-handed helical nucleoprotein filament where each Rad51 monomer binds three nucleotides. Rad51 polymerization is mediated by monomer–monomer interactions [10]. Importantly, Rad51 can also bind double-strand DNA (dsDNA) with an affinity similar to that displayed for ssDNA [11]; this binding impedes homologous pairing, and thereby HR [12]. The ssDNA/Rad51 nucleofilament (presynaptic filament) searches for a homologous template by sampling DNA in 8-nucleotides (nt) increments and capturing those tracts with 8-nt microhomology [13]. Once the ssDNA/Rad51 nucleofilament is paired with a homologous DNA sequence (synaptic complex), Rad51 promotes the strand exchange reaction in precise 3-nt steps [13,14]. The interaction of the ssDNA/Rad51 nucleofilament with the dsDNA template requires a low-affinity DNA binding site in Rad51 that is different from the high-affinity binding site that promotes the polymerization of Rad51 on ssDNA [15,16]. The strand exchange reaction generates a DNA joint molecule where Rad51 binds to the heteroduplex formed by the invading strand and its complementary strand in the donor molecule (postsynaptic complex). As a consequence of this exchange, a displacement loop (D-loop) is formed in which the invading 3′-end primes new DNA synthesis once Rad51 is disassembled by the translocase Rad54 [17]. Once this intermediate is formed, HR can proceed through different mechanisms (double-strand break repair, DSBR; synthesis-dependent strand annealing, SDSA; break-induced replication BIR) that may or may not involve gene conversion and/or crossover depending on whether there is transfer of information or reciprocal exchange of DNA between the recombining molecules, respectively (Figure 1) (for details about these mechanisms see [1,8]).

Despite its essential role in the search for homology and strand exchange, Rad51 is dispensable for some recombination events in yeast, and only a double mutant lacking Rad51 and Rad59 displays recombination defects as severe as a mutant lacking Rad52 [18]. This is due to the fact that Rad51 is not essential for BIR [19] and is not required for single-strand annealing (SSA) [20], a recombination process between direct repeats where the resected homologous sequences are directly annealed. Both Rad52 and Rad59 display strand annealing activity [21,22], which can promote strand invasion of a broken DNA end in BIR—though with low efficiency—and anneal the exposed homologous sequences in SSA [19,20]. In contrast to yeast, the lack of Rad52 in mammalian cells causes mild recombination defects. Instead, the breast and ovarian tumor suppressor protein BRCA2 maintains a central role in HR in mammals [23]. The relevance of Rad52 becomes evident in BRCA2-deficient cells [24]. The reason for these differences stems from the fact that yeast Rad52 contains both mediator (Rad51 loading onto DNA) and DNA annealing activities, which are separated in BRCA2 (mediator activity) and Rad52 (annealing activity) in mammals [25]. Yeast Rad52 mediator function is performed through physical interactions with ssDNA, Rad51, and RPA [26]. In mammalian cells, the loading of Rad51 onto the ssDNA molecule requires BRCA1, which interacts with PALB2 to recruit BRCA2. The physical interactions of BRC repeats in the middle of BRCA2 with Rad51 promoting the formation of the nucleofilament and prevent Rad51 binding to dsDNA [27,28,29]. In addition, BRCA2 stabilizes the nucleofilament through a second Rad51 interaction site in the carboxy-terminal domain (CTD) that is negatively regulated by CDK1/2-mediated phosphorylation of serine 3291 [30].

## 3. ssDNA Gap Filling by HR

A major source of DSBs occurs during S phase and is associated with the breakage of replication forks. These structures are especially vulnerable under stress conditions due to the dynamics of chromatin associated with DNA synthesis and the accumulation of ssDNA stretches and free ends [31]. Even though the mechanisms by which HR rescues broken forks remain poorly explored, BIR seems to play a major role by priming DNA synthesis via a bubble-like fork that results in Pol δ-mediated conservative replication [32,33]. Accordingly, situations that cause fork instability, such as mutations in replicative polymerases or histone depletion during S phase, activate Rad51-independent HR mechanisms [34,35].

The historical view of HR as a specialized DSB repair mechanism has been challenged by a large number of genetic and molecular evidence supporting an essential function of HR during S phase that is linked to the filling of ssDNA gaps. Genotoxic agents that stall replication forks either by generating DNA-blocking lesions (e.g., methyl methane sulfonate (MMS), UV light, 4-nitroquinoline oxide (NQO), H_2_O_2_) or reducing the pool of available dNTPs (e.g., hydroxyurea; HU, aphidicolin) uncouple the DNA unwinding and DNA synthesis activities of the replisome. This causes an accumulation of ssDNA at the fork that triggers different responses to stabilize, protect, and restart replication forks [36,37,38]. In the case of blocking lesions, the accumulation of ssDNA at the fork occurs preferentially at the leading strand, as the lagging strand can bypass the lesion by priming a new Okazaki fragment, leaving a gap behind the fork [39,40,41]. One specific consequence of the accumulation of ssDNA is the activation of the DDT response, aimed to timely complete DNA replication by promoting the bypass of the lesions and the filling of the stretches of ssDNA. This response is mediated by the ubiquitylation of the replication processivity factor PCNA (proliferating cell nuclear antigen) [42,43,44]. PCNA monoubiquitylation at lysine 164 by the heterodimer Rad6(E2)-Rad18(E3) promotes the recruitment of TLS polymerases, which provide a simple but mutagenic way to bypass the lesion and fill in the ssDNA gap at the fork (Figure 2A). Alternatively, extension of the K164 ubiquitylation with a K63-linked polyubiquitin chain by Mms2-Ubc13(E2)-Rad5 (yeast)/HLTF and SHPRH (mammal) (E3) promotes different template switching events mediated by HR proteins [7,45]. PCNA polyubiquitylation can be associated with the formation of reversed forks through a process in which the nascent strands are displaced and reannealed leading to a Holliday junction (HJ)-like structure (Figure 2B) [46]. These structures might facilitate the bypass of the blocking lesion without generating ssDNA behind the fork either by strand exchange ahead of the fork or DNA synthesis and fork restoration. Consistent with the former mechanism, yeast forks blocked by a replication fork barrier reinitiate DNA synthesis through Pol δ-mediated semiconservative replication [47]. This recombinational event might also occur without fork reversal by directly invading the intact, sister chromatid at the fork (Figure 2C), even though there are no physical evidence of this intermediate yet.

In some other cases, replication restart involves the formation of post-replicative ssDNA gaps in the daughter strand as a consequence of repriming DNA synthesis downstream of the blocking lesion (Figure 2D) [40,41]. Repriming in mammals is carried out by PrimPol, a DNA polymerase with primase activity required for progression in the presence of DNA blocking lesions [48,49]. Even though yeast cells lack a PrimPol homolog, the Pol α/primase complex seems to perform the repriming, as suggested by genetic evidence [50]. Thus, ssDNA gaps accumulate both at the leading and the lagging strand [40,41], and can be filled by TLS polymerases or processed by nucleases and helicases to generate the substrate for the HR machinery (Figure 2D) [41,51,52]. This template switching event uses most of HR central factors and generates sister chromatid junctions (SCJs) [53,54,55,56,57]. However, and in contrast to DSB-mediated HR, strand exchange during ssDNA gap filling is not initiated by the 3′-end but by the ssDNA gap through the reannealing of the parental strands, which exposes the intact newly synthesized chromatid as a template for the blocked nascent strand [58].

Both SCJs and reversed forks have been detected by electron microscopy and/or two-dimensional electrophoresis [56,58,59,60], but their abundance and relevance seem to depend on the organism and type of lesion. In mammalian cells, SCJs are hardly detected [61], whereas reversed forks are abundant structures preferentially confined to stalled forks [59,62]. In contrast, SCJs are detected in response to MMS-induced blocking lesions in yeast cells [56], whereas MMS- and UV-induced reversed forks are rare structures except in checkpoint or primase/Ctf4 mutants defective in fork stability and repriming, respectively [50,60], or in response to the topoisomerase I inhibitor camptothecin [63]. Thus, the scarcity of reversed forks in UV and MMS-treated yeast cells might reflect transient structures or a specific response to hard-to-bypass lesions. These results suggest that template switching events are predominant at the fork in mammals and behind the fork in yeast. At least for the later, this expectation is consistent with the formation of MMS- and UV-induced HR foci far away from sites of ongoing replication [64]. It must be stressed, though, that yeast Rad51 is also required at the fork as it facilitates replication fork advance in the presence of MMS-induced lesions [65,66,67] and protects nascent DNA at forks even under unperturbed conditions [41].

## 4. Non-Recombinogenic Roles of Rad51, BRCA2 and Rad52 in the Dynamics of Reversed Forks

The study of the cell response to different replication stress conditions established critical roles for Rad51 and its mediators, BRCA2 and Rad52, in the stability, protection, and restart of stalled forks. Unexpectedly, though, a number of studies along the last decade have revealed that reversed fork dynamics requires non-recombinogenic activities of these factors. The first evidence came from the study in mouse cells of stalled forks by HU, which showed that BRCA2 prevents the degradation of nascent strands by the nuclease Mre11 (Figure 2B) [68]. The analysis of *brca2* mutants demonstrated that this protective role is not mediated by the binding of BRCA2 to DNA; however, it requires its Rad51 interaction site at the CTD (eliminated in a *brca2-S3291A* mutant), suggesting that BRCA2 protects stalled forks by stabilization of the Rad51 nucleofilament. Accordingly, over-expression of a Rad51-K133R mutant protein defective in Rad51 dissociation from DNA rendered *brca2* mutant cells resistant to stalled fork degradation. Most importantly, the *brca2-S3291A* mutant was proficient in HR, indicating that stable Rad51 filaments protect stressed forks from Mre11 degradation through a non-recombinogenic function [68]. This line of research uncovered additional factors, such as BRCA1, FANCD2, RAD51C, and XRCC3, in protecting stressed forks by stabilizing Rad51 nucleofilaments [69,70]. Additional evidence for this non-recombinogenic role of Rad51 in protecting stalled forks came from the analysis of a *rad51-T131P* mutation in a patient with Fanconi anemia. The mutant protein destabilizes Rad51 nucleofilaments by constitutive activation of the Rad51 ATPase activity and renders cells sensitive to DNA interstrand crosslinks (ICLs) due to nucleolytic processing by the nuclease Dna2 and the helicase WRN; however, mutant cells remain HR proficient [71]. Importantly, a Rad51-II3A mutant protein proficient in DNA binding and nucleofilament formation but defective in strand exchange and HR protected forks arrested by a replication fork barrier in yeast or the replication inhibitor HU in human cells from nuclease degradation by Exo1 (yeast) and Mre11 (mammalian cells), further supporting a non-recombinogenic role for Rad51 in the protection of stressed forks [16,72].

Consistent with the separation of the mediator and annealing activities between BRCA2 and Rad52 in mammalian cells, yeast but not human Rad52 is required for stalled fork protection [72,73]. Indeed, mammalian Rad52, together with the PTIP complex, facilitate Mre11 targeting to stalled forks through unknown mechanisms [73]. Interestingly, the absence of either PTIP, Rad52, or Mre11 not only prevents nascent DNA degradation under stress conditions but also rescues lethality in mouse stem cells and human tumor cells defective in BRCA2, suggesting that the essential role of BRCA2 is associated with fork protection and not with HR [73,74]. However, this essential role was not observed in non-transformed human mammary epithelial cells, indicating that the contribution of HR and fork protection by Rad51 and BRCA2 is modulated by the cell context [75].

The analysis of replication intermediates by EM revealed low levels of reversed forks in *brca2* cells that were restored after eliminating PTIP, Rad52, or Mre11, indicating that the reversed fork is the entry point for the nuclease in *brca2* cells [73]; indeed, genetic conditions that prevent reversed fork formation avoid fork degradation, even though lead to elevated levels of chromosomal breakage and genetic instability [72,73]. This is, in part, due to the fact that processing reversed forks by Mre11 is necessary for their recombinational restart [76]. In this scenario, BRCA2-mediated stable Rad51 nucleofilaments would display a critical role under replication stress conditions by protecting reversed forks from excessive degradation [73,74]. This protective role, together with its DNA repair functions, through HR is critical to prevent spontaneous replication-associated DNA damage—including under-replication—that lead to mitotic abnormalities, chromosome segregation defects, and G1 arrest [75,77,78].

Studies in human cells have shown that Rad51 is not only required for the protection but also for the formation of reversed forks [59], a function that is facilitated by the RAD51B/RAD51C/RAD51D/XRCC2 subcomplex of Rad51 paralogs [79]. As previously mentioned, reversed forks form both in BRCA2-deficient and Rad51-T131P-expressing cells [73], indicating that metastable Rad51 nucleofilaments would be sufficient to promote fork reversal. However, purified human Rad51 does not have fork remodelling activity in vitro [80], suggesting that fork reversal is not mediated by the strand exchange activity of Rad51. In accordance, reversed forks form in yeast and human cells expressing the strand exchange deficient *rad51-II3A* allele [16,72]. Recent analyses in human cells has connected DDT activation with reversed fork formation through the recruitment of the translocase ZRANB3 to stalled forks by physical interactions with polyubiquitylated PCNA [61,81]. In agreement with these results, fork reversal is impeded in the absence of HLTF, leading to PrimPol and Rev1-dependent discontinuous and mutagenic replication [82]. Nevertheless, it must be stressed that the role of HLTF in fork reversal might also be associated with its DNA translocase activity [83]. Actually, a number of different DNA translocases have been shown to promote fork reversal in vitro and in vivo [3], some of which (e.g., SMARCAL1) are directed by RPA to selectively reverse forks stalled at the leading strand and restore normal forks with a lagging-strand gap [84]. In this frame, Rad51 might stimulate the fork reversal activity of these translocases (e.g., Rad54) [80]. Alternatively, it might bind to the ssDNA end of the reversed fork to stabilize the structure [2].

An immediate consequence of these studies is that the function of the Rad51 nucleofilament is associated with its stability, and that both a defect and an excess of Rad51 can lead to genomic instability. It is thereby not surprising that the role of Rad51 at stalled forks is modulated by a dynamic interplay with RPA, which in turn can regulate the activity of additional factors, such as the human translocase SMARCAL1 [2]. Mammalian cells maintain a physiological level of Rad51 by the ssDNA binding protein RADX, which counteracts BRCA2 activity by interacting with and stimulating Rad51 ATPase hydrolysis to destabilize the nucleofilament [85,86,87,88]. Strikingly, RADX-dependent Rad51 nucleofilament destabilization can either inhibit or promote fork reversal depending on whether replication forks are unperturbed or stalled, respectively. The mechanistic of this regulation might be related to the amount of ssDNA at forks; in the short tracts that accumulate at unperturbed forks, Rad51 destabilization would prevent fork reversal. In contrast, Rad51 destabilization in the long ssDNA tracts that accumulate at stalled forks would lead to metastable and dynamic filaments that would facilitate its interplay with RPA without generating roadblocks that would impair strand reannealing during fork reversal [62].

Although the strand invasion activity of Rad51 is required for efficient fork restart as determined in yeast and human cells expressing the *rad51-II3A* allele, this requirement was greater upon persistence stalling in human cells [16,72]. This has led to propose a non-recombinogenic role for Rad51 at early times of fork stalling that would favour fork restoration, whereas at late times reversed fork processing would favour fork restart by HR (Figure 2B) [16,72]. Fork restart requires the RAD51C/XRCC3 subcomplex of Rad51 paralogs but, in contrast to DSB-mediated HR, is independent of BRCA2 [74,79]. Therefore, the stability and extent of the Rad51/ssDNA nucleofilament seem to be critical for their different recombinogenic and non-recombinogenic functions.

Finally, HR plays an additional role during mitosis by promoting mitotic DNA synthesis (MiDAS) of under-replicated regions through a BIR-like mechanism that deal with DNA ends generated by nuclease digestion of stalled forks [89,90]. Interestingly, Rad51 also facilitates MiDAS at DNA regions that are not associated with DNA repair foci or γ-H2A, two DSB markers. This process is enhanced in human cells expressing Rad51-K133 and is regulated by Polo-like kinase 1, which induces Rad51 recruitment to ssDNA upon phosphorylation. These results suggest a non-recombinogenic role for Rad51 in MiDAS by protecting stalled forks against nucleases [91].

## 5. Non-Recombinogenic Roles of Rad51 and Rad52 in TLS

In contrast to HR, which is mostly error free, TLS mechanisms can be mutagenic as most TLS polymerases display low fidelity. In *Saccharomyces cerevisiae*, there are three TLS polymerases that operate with different affinities depending on the dose and type of blocking lesion: Rev1, Pol ζ (formed by the catalytic subunit Rev3 and the regulatory subunits Rev7, Pol31, and Pol32), and Pol η (encoded by *RAD30*) [7]. Their activity—at least for Rev1/Pol ζ—is most dominant in G2/M [92], where they compete with a UbPCNA-independent salvage pathway of HR (inhibited during S phase) to deal with DNA gaps left unrepaired by the main, UbPCNA-dependent mechanism of HR [45]. This salvage pathway, in contrast to the latter, can lead to chromosomal rearrangements [93]. Therefore, the choice of the DDT mechanism is critical for the maintenance of genome integrity.

The analysis of DNA damage-induced repair foci in yeast revealed the requirement of the TLS machinery (Rad6/Rad18-mediated PCNA ubiquitylation and polymerases Rev1/Pol ζ) for the resolution of Rad52 but not Rad54-associated foci, indicating that this phenotype was not a consequence of HR being the only operative ssDNA filling process. Likewise, persistence of MMS-induced Rad52 foci in TLS mutants was not due to a demand for TLS polymerases during MMS or UV-induced HR, but a consequence of yeast Rad52 acting in concert with the TLS machinery to fill in the stretches of ssDNA generated during DDT in response to these genotoxic agents. In accordance, yeast Rad52 is partially required for MMS- and UV-induced mutagenesis [5]. This result was unexpected as inferred by epistatic analyses of DNA damage sensitivity and mutagenesis between HR and TLS mutants [94,95,96]. This synergism is explained by the fact that DNA damage-induced TLS is still partially operative in the *rad52∆* mutant [5]. Yeast Rad52 is shown to facilitate Rad6/Rad18 chromatin binding and, in response to DNA damage, PCNA ubiquitylation [5]. Importantly, this function was not observed in the absence of Rad54, which is essential for HR during DDT [5,54,97,98]. This means that Rad52 facilitates Rad6/Rad18 recruitment to chromatin and subsequent PCNA ubiquitylation through non-recombinogenic activities. However, Rad51 and Rad57 cooperate with Rad52 in this mechanism but are dispensable for TLS-mediated ssDNA filling and mutagenesis, suggesting that Rad6/Rad18 recruitment to chromatin alone cannot explain the TLS defects observed in the *rad52∆* mutant [5].

Rad6/Rad18 is targeted to chromatin by Rad18 binding to ssDNA, RPA (yeast and mammal), and sumoylated PCNA (only yeast) [44,99,100,101,102]. The involvement of Rad52 and Rad57 in the recruitment of Rad6/Rad18 suggests that this function is mediated by the Rad51 nucleofilament. Therefore, RPA and Rad51 are likely cooperating to target Rad6/Rad18 and promote PCNA ubiquitylation in response to DNA damage (Figure 2D; note that this event might also occur at the fork). The competition between RPA and Rad51 for ssDNA binding would explain the mild defects in TLS observed in the HR mutants [5]. Remarkably, the Rad6/Rad18 recruitment to chromatin is observed even in the absence of DNA damage [5]. Therefore, these interactions might occur at not-yet defined regions and, in response to DNA damage, mobilize Rad6/Rad18 to ssDNA gaps to ubiquitylate PCNA. This ubiquitylation would facilitate template switching events in S phase and TLS in G2/M. Some aspects of this mechanism seem to be conserved in human cells, where Rad51 physically interact with Rad18 and FANCD2 in a complex that is stimulated specifically by HU. In response to this agent, these factors promote PCNA monoubiquitylation and chromatin recruitment of the TLS polymerase Pol H; importantly, these events are not affected by the absence of BRCA2 or the pharmacological inhibition of Rad51, indicating that they are independent of HR [4].

Another piece of evidence supporting the non-recombinogenic role of Rad51 and Rad52 in the DDT response comes from the analysis in *S. cerevisiae* of the physical interactions between these proteins and the MCM complex, an essential component of the CMG (Cdc45/MCM/GINS) replicative helicase [6]. A *rad51* mutation that disrupts the interaction between Rad51 and MCM was proficient in MMS and DSB-induced HR but partially defective in MMS-induced ssDNA gap filling, supporting a role for this interaction in TLS. Actually, this mutant was also partially defective in replication fork progression through damaged DNA and this defect could be bypassed by forcing the interaction through the simultaneous expression of Mcm4-GFP and Rad51-GBP (GFP-binding protein) chimeras. Thus, the MCM–Rad51 interaction facilitates DNA damage-induced ssDNA gap filling and fork progression through DNA blocking lesions by non-recombinogenic functions, presumably TLS (Figure 2A,D) [6].

How does the MCM–Rad51 interaction promote replicative and repair functions during yeast DDT? The biochemical characterization of these interactions revealed interesting aspects about its regulation. First, MCM also interacts with Rad52, but Rad52 does not bridge the MCM–Rad51 interaction. Second, these interactions occur in G1 and are lost in S phase, unless cells are released in the presence of replicative stress. Third, these interactions are prevented at origins and replication forks; instead, they occur at a cellular fraction that, in contrast to most chromatin, is insoluble after DNA digestion with nucleases. Rad51 accumulates in a MCM- and DNA binding-independent manner, whereas MCM seems to bind DNA. In this nucleoprotein scaffold, MCM and Rad51 (and also Rad52) display dynamic interactions that are regulated by the kinase activity of Cdc7, which prevents the release of Rad51 and Rad52 from the insoluble scaffold in response to replicative stress [6]. One possibility is that these physical interactions provide a platform to facilitate the chromatin recruitment of Rad6/Rad18 by Rad51 and Rad52. PCNA ubiquitylation by Rad6/Rad18 would promote the targeting of TLS polymerases to fill in ssDNA gaps [103] and would facilitate replication fork advance in the presence of damaged DNA [104], mechanistically connecting the two functions reported for the MCM–Rad51 interaction. In this regard, it is worth noting that Rad51-dependent PCNA monoubiquitylation in human cells is not affected in the absence of BRCA2, which prevents Rad51 binding to chromatin [4]. However, yeast Rad51 and Rad52 promote Rad6/Rad18 binding to DNA also under unperturbed replication conditions [5] that do not sustain the MCM–Rad51–Rad52 interactions [6], suggesting that they could also operate through alternative mechanisms.

Remarkably, this nuclease insoluble fraction also accumulates a number of factors involved in replication fork stability: the topoisomerase Top2, the checkpoint effector Rad53, the helicase Sgs1, the origin recognition complex (ORC) and the Cdc7 regulator Dbf4 [105,106]. We have hypothesized that these factors, together with MCM, Rad51, and Rad52, aggregate in G1 for replication assistance [6]. Under unperturbed replication conditions, Rad51 and Rad52 (and may be others) have to be removed to prevent toxic protein–DNA interactions [107,108]; however, in response to replicative DNA damage, Cdc7 would maintain these physical interactions to assist stressed forks by facilitating their advance and the repair of ssDNA gaps generated during lesion bypass [6]. This function might be conserved in mammalian cells, where the MCM–Rad51–Rad52 interactions have also been observed [109,110]. In this regard, it worth noting that mammalian replication origins and components of the pre-replication complex, including MCM, are associated during G1 and early S phase with a nuclease-insoluble scaffold, where DNA would be replicated at static replisomes [111,112].

## 6. Concluding Remarks

Thus far, the role of Rad51, BRCA2, and Rad52 in DDT was restricted to HR mechanisms. A new scenario has emerged in which they additionally operate in DDT through non-recombinogenic functions. A major conclusion of these results is that Rad51 and its mediators take part in most DDT mechanisms, and therefore they become a potential molecular switch to decide how to tolerate the lesion at different stages. At the fork, the Rad51 nucleofilament is critical for the formation, protection, and restart of reversed forks [2]. The involvement of Rad51 and Rad52 in TLS and fork advance through damaged DNA opens up the possibility that they could also facilitate fork restart by this mutagenic pathway [4,5,6]. Indeed, RecA can promote the switch from the replicative to a TLS polymerase in the bacterial replisome [113]. Interestingly, the analysis of stressed forks in *Xenopus laevis* showed that Rad51, apart from promoting fork reversion in a BRCA2-independent manner, interacts and stabilizes Pol α, and therefore might play a role in DNA synthesis resumption and/or repriming [114]. Behind the fork, yeast Rad51 and Rad52 may channel the filling of unrepaired ssDNA gaps in G2/M, either by TLS (via Rad6/Rad18 recruitment and not-yet defined Rad52 functions) or HR (via DNA strand exchange) [5]. Since the consequences for genome integrity are different depending on the mechanism of fork restart and ssDNA filling [115], it will be important to gain a deeper insight into these novel non-recombinogenic functions to understand how Rad51 and its mediators are regulated in response to the type and extent of DNA lesions and the cell cycle phase.

## Figures and Tables

**Figure 1 genes-12-01550-f001:**
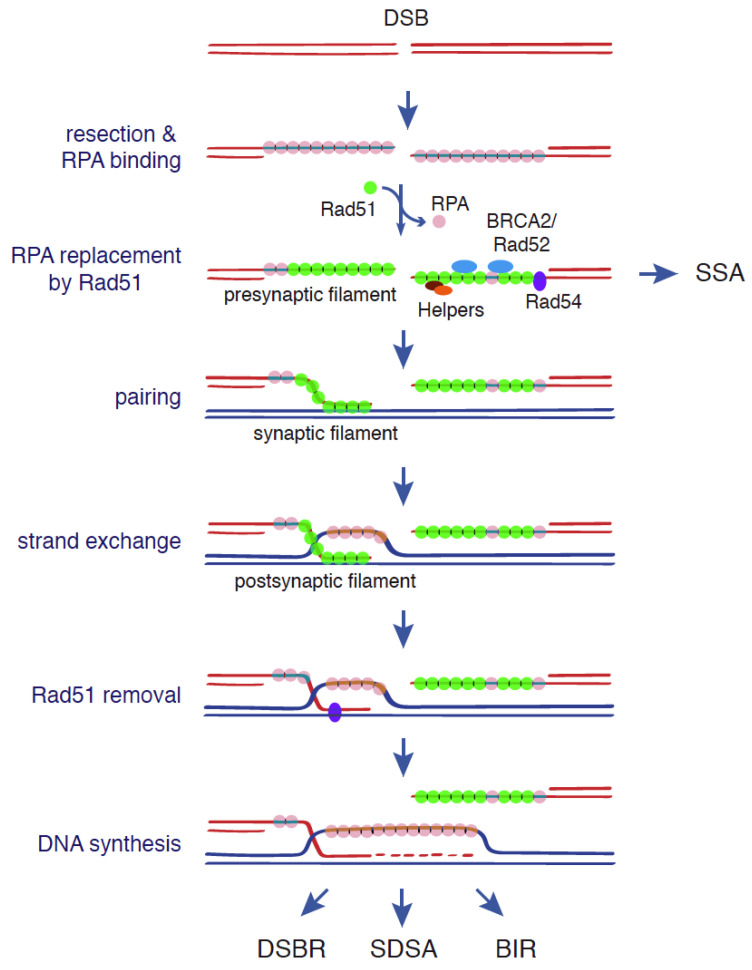
Canonical functions of Rad51, BRCA2, and Rad52 during DSB repair. The central steps in DSB-induced HR are located for homology and strand exchange. They are carried out by a Rad51 nucleofilament formed upon resection of the 5′-ends of the DSB and subsequent replacement of the ssDNA binding complex RPA with Rad51. This replacement is mediated by BRCA2 in mammalian cells and Rad52 in yeast, with the help of accessory factors. DNA strand exchange leads to the formation of a D-loop structure that is enlarged by DNA synthesis once Rad54 removes Rad51. The final output of the repair will depend on the chosen mechanism: DSBR, SDSA, BIR, or SSA [1,8]. See text for more details.

**Figure 2 genes-12-01550-f002:**
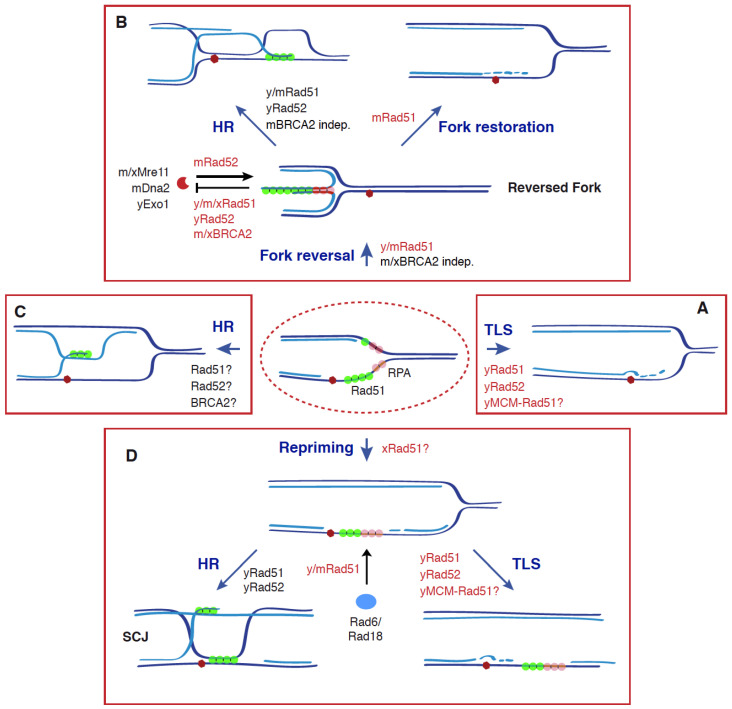
Recombinogenic and non-recombinogenic functions of Rad51, BRCA2, and Rad52 in DDT. The DDT response is triggered by an accumulation of ssDNA at the fork as a consequence of stress conditions (e.g., blocking lesions, dNTP depletion) that uncouple DNA unwinding and DNA synthesis activities. This response is characterized by ubiquitylation of PCNA, which facilitates the bypass of the blocking lesion and the fill in of the ssDNA gaps through different mechanisms: TLS (**A**), fork reversal, followed by HR or fork restoration (**B**), HR (**C**), and repriming, followed by HR or TLS (**D**). Apart from its canonical role in HR by promoting strand exchange, Rad51 plays additional, non-recombinogenic roles together with BRCA2 and Rad52 in (1) the formation, protection, and restart of reversed forks through Rad51 nucleofilaments; (2) TLS, by facilitating Rad6/Rad18 recruitment to chromatin and subsequent PCNA ubiquitylation, and by physically interacting with the helicase MCM at nucleoprotein scaffolds. All these functions make Rad51 a potential molecular switch to choose the DDT mechanism. See text for more details. Dark and light blue indicate parental and nascent strands; a red hexagon indicates a blocking DNA lesion; non-recombinogenic functions of Rad51, Rad52, and BRCA2 are highlighted in red; y (yeast), m (mammal), and x (Xenopus) indicate the species in which those molecular functions have been shown for each protein; a question mark indicates a putative function.

## Data Availability

Not applicable.

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
