# Peer review of "Non-Recombinogenic Functions of Rad51, BRCA2, and Rad52 in DNA Damage Tolerance"

_genes, 2021, doi:10.3390/genes12101550_

Round 1
Reviewer 1 Report
In this review, the author describes the role of Rad51, BRCA2 and Rad52 in DNA damage tolerance. While these proteins have been known to play a role in homologous recombination at double-strand breaks and single-strand gap, it has recently been evidenced that they also play non-recombinogenic functions. After a nice introduction, the review exposes the role of HR first during double strand break repair, then in ssDNA gap repair, which are the canonical views of HR. The review focuses then on the non-recombinogenic roles of Rad51, BRCA2 and Rad52 in lesion bypass. This latest part focuses mainly on the two recent papers from the author's lab.
The review is well written and interesting to read. It gives a nice overview of the different processes related to DNA damage tolerance, and the role of the several proteins involved both in yeast and in mammalian cells.
Main comments:
-It is very interesting and useful to do the parallel between the yeast and mammalian systems. However, in some places in the manuscript, it is not clear to which organism the author refers to, which can be confusing. This should be clarified all along the manuscript. Using hRad52 or yRad52 for instance could avoid confusion in some places.
-The "RF" abbreviation is used by the author for "reversed fork". As the same abbreviation has also been commonly used for "Replication Fork" this could be confusing for the reader. I would recommend not using use the abbreviation for this term.
Minor points
P1 line 34: "in more details"
P2, lines 87 and 90: the references are not formatted with numbers.
P4, line 153: It feels more logical to write the sentence as follow: "Even though yeast cells lack a PrimPol homolog, the Pol alpha/primase complex…"
P5, lines 199-203: it would have been nice to describe what has been observed with the rad51-II3A mutant in the two cited references.
P6, line 263: "select" might be misleading as Polymerases are not really selected per se, but it is rather a trial and error that defines which Polymerase will do the insertion.
P6, Line 266: I feel that "compete" will be more appropriate than "cooperate"
P7, lines 290-: The interaction of Rad18 with RPA, Ub-PCNA has been described in yeast (3 cited references). Has this been observed in mammalian cells as well?
P7, line 323: the "nuclease insoluble fraction" should be explained in more detail
Reference 70: the first author's last name is "Ait Saada"
Figure 2 is too small and the resolution is not good enough (shows pixels when enlarged).
Author Response
In this review, the author describes the role of Rad51, BRCA2 and Rad52 in DNA damage tolerance. While these proteins have been known to play a role in homologous recombination at double-strand breaks and single-strand gap, it has recently been evidenced that they also play non-recombinogenic functions. After a nice introduction, the review exposes the role of HR first during double strand break repair, then in ssDNA gap repair, which are the canonical views of HR. The review focuses then on the non-recombinogenic roles of Rad51, BRCA2 and Rad52 in lesion bypass. This latest part focuses mainly on the two recent papers from the author's lab.
The review is well written and interesting to read. It gives a nice overview of the different processes related to DNA damage tolerance, and the role of the several proteins involved both in yeast and in mammalian cells.
Main comments:
-It is very interesting and useful to do the parallel between the yeast and mammalian systems. However, in some places in the manuscript, it is not clear to which organism the author refers to, which can be confusing. This should be clarified all along the manuscript. Using hRad52 or yRad52 for instance could avoid confusion in some places.
We prefer not to follow that nomenclature because in many cases the reference refers to a general feature of the protein and in some others to mammalian, non-human cells or even Xenopus, and the text also contains many other proteins whose knowledge comes from different organisms. We have revised the text to specifically mention the cell origin when it is important to compare between organisms. We have followed the reviewer’s suggestion in Figure 2 to clarify in what organisms those specific functions have been reported.
-The "RF" abbreviation is used by the author for "reversed fork". As the same abbreviation has also been commonly used for "Replication Fork" this could be confusing for the reader. I would recommend not using use the abbreviation for this term.
Done as suggested
Minor points
P1 line 34: "in more details"
Corrected
P2, lines 87 and 90: the references are not formatted with numbers.
Corrected
P4, line 153: It feels more logical to write the sentence as follow: "Even though yeast cells lack a PrimPol homolog, the Pol alpha/primase complex…"
Changed as suggested
P5, lines 199-203: it would have been nice to describe what has been observed with the rad51-II3A mutant in the two cited references.
The sentence has been rephrased to first indicate their observations and then the conclusion from those results.
P6, line 263: "select" might be misleading as Polymerases are not really selected per se, but it is rather a trial and error that defines which Polymerase will do the insertion.
Corrected; now it reads “there are three TLS polymerases that operate with different affinities depending on the dose and type of blocking lesion”.
P6, Line 266: I feel that "compete" will be more appropriate than "cooperate"
Changed as suggested
P7, lines 290-: The interaction of Rad18 with RPA, Ub-PCNA has been described in yeast (3 cited references). Has this been observed in mammalian cells as well?
Indeed it was. Human Rad18 has ssDNA binding activity and interacts with RPA, but not with Sumoylated PCNA. This information and the missing reference has now been included.
P7, line 323: the "nuclease insoluble fraction" should be explained in more detail
Further information has been included about this fraction. Nevertheless, note that little is known yet about the molecular nature of this compartment.
Reference 70: the first author's last name is "Ait Saada"
Corrected
Figure 2 is too small and the resolution is not good enough (shows pixels when enlarged).
Editors have been informed to enlarge the Figure and get a better quality file.
Reviewer 2 Report
This manuscript is comprehensive, demonstrating deep knowledge of the role of homologous recombination (HR) and HR-proteins in protecting and re-starting stalled replication forks. As the title indicates, the most interesting part of the review focus on when Rad51, Rad52 and BRCA2 displays functions that appear to be non-recombinogenic, i.e. when their role is independent of the strand exchange activity of Rad51.
I have three concerns:
1. The manuscript would benefit from differentiating between the leading and lagging strands during replication and if the different repair mechanism described act preferentially on either of them.
2. The manuscript would benefit from emphasizing the latter parts (i.e. sections 4 and 5) and perhaps shortening the earlier parts (2 and 3), since it is the non-recombinogenic roles of HR-proteins that is novel and interesting.
3. Figure 2 would be improved with a graphical legend explaining different colors used and highlighting the potential non-recombinogenic roles of BRCA2 and Rad51/52 (why is BRCA2 crossed out etc.). The figure is also small and it is hence different to distinguish all details. In summary, both the figure and figure legend need improvement.
Minor comments:
34 A strange sentence as human cells are not really considered a model organism and the verb ending the sentence has the wrong tense.
182 have not has
309 Would it be more correct to name the replicative helicase the CMG-complex (Cdc45/Mcm2-7/GINS)?
365 chose or decide on instead of choice
Author Response
This manuscript is comprehensive, demonstrating deep knowledge of the role of homologous recombination (HR) and HR-proteins in protecting and re-starting stalled replication forks. As the title indicates, the most interesting part of the review focus on when Rad51, Rad52 and BRCA2 displays functions that appear to be non-recombinogenic, i.e. when their role is independent of the strand exchange activity of Rad51.
I have three concerns:
- The manuscript would benefit from differentiating between the leading and lagging strands during replication and if the different repair mechanism described act preferentially on either of them.
I agree that the location of the lesion on the leading versus the lagging strand is an important factor regulating DDT mechanisms. This has been stressed now by explaining how a blocking lesion in the leading or the lagging affects the accumulation of ssDNA, and the importance of this accumulation in the selection of the translocases that operate at reversed forks. Little is known, though, about how a post-replicative gap in the leading or the lagging strand can affect the mechanism of DDT. The absence of both Rad52 and Rad5 in yeast led to a synergistic defect in MMS-induced gap filling that was interpreted as the consequence of being operating in the lagging and leading, respectively (Gangavarapu et al 2007). However, this model is so far speculative and not supported by further evidence. We have not extended this issue longer because there are no studies that far that link it with the non-recombinogenic functions of the HR proteins, apart from the translocase selectivity.
- The manuscript would benefit from emphasizing the latter parts (i.e. sections 4 and 5) and perhaps shortening the earlier parts (2 and 3), since it is the non-recombinogenic roles of HR-proteins that is novel and interesting.
Sections 2 and 3 are restricted to the minimal information for a non-expert reader to understand the non-recombinogenic roles of HR proteins, and we prefer not to short it to facilitate the reading of the following sections. Section 4 is focused on the biology of reversed forks. There are excellent and more extensive reviews by the people that directly worked on this topic (e. g., Bert et al. 2020, Nat Rev Mol Cell Biol) and we therefore limited this section to those aspects directly connected with non-recombinogenic function of HR proteins. Nevertheless, we have extended this section following Reviewer 3 suggestions to include and discuss some studies about the mitotic consequences of the absence of recombinogenic and non-recombinogenic functions of HR proteins, as well a missing information about non-recombinogenic roles of Rad51 in MiDAS. Last section review our last two papers, where we describe for the first time non-recombinogenic functions of Rad51 and Rad52 in TLS. We prefer not to go in more details in this section – that are in the original papers – and discuss the main conclusions with previous data in a more general way.
- Figure 2 would be improved with a graphical legend explaining different colors used and highlighting the potential non-recombinogenic roles of BRCA2 and Rad51/52 (why is BRCA2 crossed out etc.). The figure is also small and it is hence different to distinguish all details. In summary, both the figure and figure legend need improvement.
Editors have been informed to enlarge the Figure and get a better quality file. In addition, it has been modified to include further information concerning the species where the molecular functions have been reported. Colors and symbols have been explained as suggested.
Minor comments:
34 A strange sentence as human cells are not really considered a model organism and the verb ending the sentence has the wrong tense.
Corrected
182 have not has
Corrected
309 Would it be more correct to name the replicative helicase the CMG-complex (Cdc45/Mcm2-7/GINS)?
Changed as suggested; now it reads:” Another piece of evidence supporting a non-recombinogenic role of Rad51 and Rad52 in the DDT response comes from the analysis in S. cerevisiae of the physical interactions between these proteins and the MCM complex, an essential component of the CMG (Cdc45/MCM/GINS) replicative helicase.”
365 chose or decide on instead of choice
Corrected
Reviewer 3 Report
This review article of Félix Prado on RAD51 pathway functions that are independent of strand invasion and recombination is well-written and contains a lot of interesting interpretations and models on the current state of the field. Below are a few minor comments. I recommend the publication of this review after minor editing.
Minor comments.
- The role of BRCA-RAD51 pathway in fork protection and fork regression is well-described in the manuscript. However, a short discussion of the works of Bernard Lopes (Wilhelm et al. PNAS 2014, PMID: 24347643) and Madalena Tarsounas (Lai et al. Nat com 2017 PMID: 28714477) on the impact of BRCA2 and RAD51 deficiencies on fork progression are lacking. Both labs have shown by fiber analysis that fork speed is reduced in either BRCA2 or RAD51 deficient mammalian cells. It would be interesting to tell to the readers that the measured fork progression can be impacted by cycles of fork pausing, reversion, degradation of nascent strands and fork restart (or alternative models). This is consistent with the recent literature showing in diverse contexts that fork progression is impacted by fork reversal, translesion synthesis and repriming activities.
- The Esashi Lab has just published a nice paper showing the involvement of RAD51 in MIDAS. These new results should be included in the manuscript (Wassing et al. Nat com 2021, PMID: 34508092). Again these recent results are in line with findings of Bernard Lopes (Wilhelm et al. PNAS 2014, PMID: 24347643) and Maria Jasin showing that BRCA-deficient cells suffer from mitotic catastrophes (Feng and Jasin Nat com 2017, PMID: 28904335). These findings should be re-contextualized to emphasize on non-recombinogenic functions of the BRCA-RAD51 pathway that impact on mitosis (ssDNA gaps, under-replication, MIDAS…).
- Figure 2 is too small on the current layout of the manuscript.
Author Response
This review article of Félix Prado on RAD51 pathway functions that are independent of strand invasion and recombination is well-written and contains a lot of interesting interpretations and models on the current state of the field. Below are a few minor comments. I recommend the publication of this review after minor editing.
Minor comments.
- The role of BRCA-RAD51 pathway in fork protection and fork regression is well-described in the manuscript. However, a short discussion of the works of Bernard Lopes (Wilhelm et al. PNAS 2014, PMID: 24347643) and Madalena Tarsounas (Lai et al. Nat com 2017 PMID: 28714477) on the impact of BRCA2 and RAD51 deficiencies on fork progression are lacking. Both labs have shown by fiber analysis that fork speed is reduced in either BRCA2 or RAD51 deficient mammalian cells. It would be interesting to tell to the readers that the measured fork progression can be impacted by cycles of fork pausing, reversion, degradation of nascent strands and fork restart (or alternative models). This is consistent with the recent literature showing in diverse contexts that fork progression is impacted by fork reversal, translesion synthesis and repriming activities.
As mentioned by the reviewer, BRCA2 or RAD51 deficient mammalian cells display slow replication (Daboussi et al JCS 2008; Lai et al. Nat Comm 2017; Wilhelm et al. PNAS 2014). However, this slow replication phenotype is observed under unperturbed conditions. In response to DNA damage, DNA replication is faster in mammalian and DT40 cells deficient in HR than in wild type cells (Henry-Mowatt et al Mol Cell 2003; Zellweger et al JCB 2015). In contrast to higher eukaryotes, DNA replication is slowed down in yeast mutants defective in HR both under unperturbed and stressed conditions (Vazquez et al DNA repair 2008; Alabert et al EMBO J 2009; Gonzalez-Prieto EMBO J 2013). These differences may reflect the nature of the replication obstacles and/or different mechanisms of HR, but so far this is unknown. Thus, we prefer not to enter into this topic that goes beyond the scope of the review.
The Esashi Lab has just published a nice paper showing the involvement of RAD51 in MIDAS. These new results should be included in the manuscript (Wassing et al. Nat com 2021, PMID: 34508092). Again these recent results are in line with findings of Bernard Lopes (Wilhelm et al. PNAS 2014, PMID: 24347643) and Maria Jasin showing that BRCA-deficient cells suffer from mitotic catastrophes (Feng and Jasin Nat com 2017, PMID: 28904335). These findings should be re-contextualized to emphasize on non-recombinogenic functions of the BRCA-RAD51 pathway that impact on mitosis (ssDNA gaps, under-replication, MIDAS…).
This is an interesting topic that is also analyzed in the papers mentioned in the first point. We have discussed it in the context of the genetic consequences of the lack of Rad51 and BRCA2. Now it reads in section 4: “In this scenario, BRCA2-mediated stable Rad51 nucleofilaments would display a critical role under replication stress conditions by protecting reversed forks from excessive degradation [74,75]. This protective role, together with its DNA repair functions through HR is critical to prevent spontaneous replication-associated DNA damage – including under-replication – that lead to mitotic abnormalities, chromosome segregation defects and G1 arrest [76,78,79].”
We also appreciate the reviewer’s reference to the work by Eshadi’s Lab, as they describe an important non-recombinogenic role of Rad51 in the process of MiDAS. This has also been discussed at the end of section 4: “Finally, HR plays an additional role during mitosis by promoting mitotic DNA synthesis (MiDAS) of under-replicated regions through a BIR-like mechanism that deal with DNA ends generated by nuclease digestion of stalled forks [89,90]. Interestingly, Rad51 also facilitates MiDAS at DNA regions that are not associated with DNA repair foci or g-H2A, two DSB markers. This process is enhanced in human cells expressing Rad51-K133 and is regulated by Polo-like kinase 1, which induces Rad51 recruitment to ssDNA upon phosphorylation. These results suggest a non-recombinogenic role for Rad51 in MiDAS by protecting stalled forks against nucleases [91].”
Finally, we have discussed an interesting observation discussed by Feng and Jasin (Nat Commun 2017) about the essentiality of Rad51 and BRCA2. Now it reads: “Interestingly, the absence of either PTIP, Rad52 or Mre11 not only prevents nascent DNA degradation under stress conditions but also rescues lethality in mouse stem cells and human tumor cells defective in BRCA2, suggesting that the essential role of BRCA2 is associated with fork protection and not with HR [74,75]. However, this essential role was not observed in non-transformed human mammary epithelial cells, suggesting that the contribution of HR and fork protection by Rad51 and BRCA2 is modulated by the cell context [76].”
- Figure 2 is too small on the current layout of the manuscript.
Editors have been informed to enlarge the Figure and get a better quality file.